# The Impact of the Urban Built Environment on the Play Behavior of Children with ASD

**DOI:** 10.3390/ijerph192214752

**Published:** 2022-11-10

**Authors:** Shengzhen Wu, Chen Pan, Lihao Yao, Xiaojing Wu

**Affiliations:** 1College of Arts and Design, Jimei University, Xiamen 361000, China; 2School of Art and Design, Guangdong University of Petrochemical Technology, Maoming 525000, China; 3College of Coastal Agricultural Sciences, Guangdong Ocean University, Zhanjiang 524000, China; 4School of Architecture and Urban Planning, Zhuhai College of Science and Technology, Zhuhai 519041, China; 5Department of Economic Management, Fujian Economic and Trade School, Quanzhou 362000, China

**Keywords:** children with ASD, social anxiety, play behavior, urban built environment

## Abstract

Anxiety caused by the lack of social skills is the biggest problem faced by children with ASD. Playing can improve children’s social skills and relieve anxiety. This study aimed to explore the influence of urban built environments on ASD children’s play behavior. The participants in this study were 57 parents of children with ASD. An anonymous questionnaire was used to collect and analyze data. At the same time, retrospective semi-structured interviews with 31 parents of ASD children were performed to validate the data analysis results. The results showed that lower residential building density, higher residential greening and higher destination accessibility have positive effects on ASD children’s play behavior. Excellent transportation facilities and high NDVI vegetation coverage have positive effects on the play behavior of children with ASD. More recreational facilities and recreational playability have positive impacts on the play behavior of children with ASD. The population density and number of children in the destination, as well as public facilities, influence the play behavior of children with ASD. The research results can promote the integration of this group into urban life and further promote social equity. At the same time, with the social needs of autistic children as an intermediary, it is expected to further explore new directions for sustainable urban development. Finally, combined with the research results, parents of ASD children are given proposals for how to increase the likelihood of children’s play behavior by choosing appropriate urban built environments.

## 1. Introduction

According to the latest autism spectrum disorder statistics released by the US Centers for Disease Control and Prevention in 2020, the incidence of autistic spectrum disorder children (hereinafter referred to as ASD children) in the United States reached as high as 1:54, an increase of nearly 10% compared with the results of the 2018 statistics (1:59) [1]. At the same time, according to the “Report on the Development of China’s Autism Education and Rehabilitation Industry III” (2019), the number of ASD children aged 0 to 6 in China exceeds 2 million. A core feature of autism is severe deficits in the acquisition and use of social skills and abilities [2], and these deficits can affect children’s social interactions, which in turn evolve into anxiety. Play is a manifestation of early childhood social behavior, and children practice social skills with their peers through games [3]. Montessori education uses a play-based educational approach to effectively help children with ASD develop the social skills they need, to change their behavior and promote their own development [4]. Therefore, play behavior is crucial to the promotion or maintenance of social skills in children with ASD. Lynch pointed out in The Image of the City that children perceive the urban environment through self-experience, and the general process of interest is arousal-observation-selective participation-social behavior-forming an overall impression of the urban environment [5]. Marcus also discussed children’s communication behaviors in People Places: Design Guidelines for Urban Open Space, and divided children’s play behaviors in the urban built environment: observation-participation-avoidance-concealment [6]. However, due to regional differences, urban built environment impacts vary widely between developed and developing countries [7], and due to differences in cultural backgrounds, diagnoses of children with ASD in other countries may not provide cross-culturally valid results [8]. Therefore, this study uses the Autism Diagnostic Scale (Table 1), which is suitable for China’s national conditions, as a reference and combines the Autism Children’s Development Assessment Scale (Table 2) issued by the China Disabled Persons’ Federation to describe the play behavior of children with ASD.

In past research, social impairments in ASD children were thought to be directly related to their personal deficits. However, a growing body of research has found that environmental factors influence their interactions with the social environment [9]. Therefore, exploring the built environment will be able to effectively promote play behavior and alleviate social anxiety in children with ASD through planning and design. 

The urban built environment refers to manmade environments built to serve human activities, including large-scale urban environments, excluding the environment formed by nature. The earliest public transport-oriented development model proposed the 3D theory of urban built environment, where D refers to density, diversity and design [10]. Subsequently, the 5D theory was proposed in the walking-oriented micro-city design, expanding to include distance and destination accessibility [11]. The essence of play is activity, and almost all the main activities of children in daily life are related to games. Research has demonstrated that urban built environments can influence children’s activities [12]. For instance, moderate urban density is friendly to children. Residential density and population density were positively correlated with the possibility of children’s independent activities, while building density was negatively correlated with the possibility of children’s independent activities [13]. Meanwhile, the density of sports venues, activity facilities and street intersections around residential areas have significant positive correlations with children’s moderate- and high-intensity physical activity [14]. The availability of parkland and the presence of sidewalks are both environmental factors that positively promote children’s play activities [15]. In a study exploring design strategies for outdoor play environments for children with ASD, setting up collaborative and mobile recreational facilities can improve the social skills of children with ASD [16]. The physical environment factors that hinder children with ASD from participating in play are excessively harsh lighting, unusual noise, crowds and crowded spaces [17]. Imperviousness suggests that both mulch and canopy cover increase the risk of severe anxiety in children with ASD, and children with autism may experience the benefits of nature’s stress relief differently than their normally developing peers [18]. 

Cities are designed for the general public. This study investigated the impact of the urban built environment on the play behavior of children with ASD. The research results can promote the integration of this group into urban life and further promote social equity. At the same time, with the social needs of autistic children as an intermediary, it is expected to further explore new directions for sustainable urban development. In the past 20 years, the research on the relationship between international children and youth activities and the built environment has focused on the impact of urban community buildings and planning environment on youth physical activity. Therefore, this study is expected to enrich the related theories on the built environment and ASD children’s play behavior, provide a new vision for building a child-friendly city and create a healthier and more inclusive urban environment for other socially disadvantaged groups. 

Two experimental methods were used in the study, anonymous questionnaires and retrospective semi-structured interview. The reason for using two data collection methods for the study is, on the one hand, to objectively quantify the play behavior of children with ASD. On the other hand, because of the particularity of ASD children, their play behavior is uncertain. Two data collection methods can maximize the accuracy and reliability of the research results.

## 2. Methodology

The sampling site of this study was the special education school in Jinjiang City, Fujian Province (Figure 1). The school is the only nine-year compulsory education school in Quanzhou that specializes in autism, and it is also the school with the largest number of ASD children among the special education schools in the three cities of Xiamen, Zhangzhou and Quanzhou. Before the start of the study, we tried to contact the special education school personnel in Xiamen and Zhangzhou and invited them to assist in the study. However, due to the low level of cooperation of parents of children with ASD, the questionnaires collected in these two cities are almost absent. Therefore, in order to eliminate the experimental error caused by regional factors, the sampling location is set as the special education school in Jinjiang City, Fujian Province. 

### 2.1. Playing Behavior of Children with ASD

Most children with ASD have language barriers, making it difficult for them to understand and respond to nonverbal behaviors [19]. Therefore, language may be meaningless, and it is difficult to extract useful data and make objective judgments in the later stage. For this reason, we conducted interviews with teachers at special education schools and psychological clinicians, and the general steps they agreed on regarding the play behavior of children with ASD applicable to this study are shown in Table 3. 

### 2.2. Urban Built Environment

According to random interviews with parents of ASD children in the early stage, due to the particularity of the children themselves, they are generally accompanied by guardians, the main way of travel is walking, and most of the travel destinations are urban green spaces that can be reached on foot near the residence. In order to exclude subjective factors as much as possible, this study used WeChat mini program with positioning function instead of traditional questionnaire tools to collect urban built environment data. Parents were required to log on the WeChat mini-program and keep it running while going out. Then, after arriving at the destination, observe the play behavior of the ASD children and complete the questionnaire. At the same time, the mini program recorded coordinates every 5 s. In the server background, the coordinates of each questionnaire were imported into GIS (geographic information system) to generate travel routes. According to the 10 min walking distance of children in the existing literature, the radius of the travel path was determined to be 500 m. Taking this as a reference, the quantitative information on the path area was extracted and sorted, and the original data were obtained for subsequent analysis. The detailed steps are shown in Figure 2. 

WeChat is currently the social software with the largest number of users in China, with a total of 1.26 billion users so far. The WeChat mini program is an application that can be used without downloading and installing. Compared with the previous use of GPS positioning equipment for children to obtain positioning information, the WeChat mini program has the following advantages:(1)Solves the problem that the device needs to be charged regularly for battery life. In the previous study, the staff had to charge the equipment and export the data regularly every day.(2)Completely removes the interference to children of the device itself. The mini program is operated by parents, and they fill in questionnaires; the whole process does not have any on-device impact on children with autism.(3)Avoids unreasonable data. The mini program guides the user, accurately prompting how to operate in each step; if the operation is not followed, the next step cannot be entered.(4)The data export results are clearer. The completion of the questionnaire can be monitored in real time through the backend of the mini program, and the exported data items are clear and easy to understand.

The urban built environment data involved in this study are: residential building density, residential greening, destination accessibility, transportation service facilities, NDVI vegetation coverage, population density, number of children in the destination, public facilities, number of recreational facilities and recreational playability. 

The data sources for the quantitative indicators of the urban built environment are as follows: The population density data come from the 2020 Jinjiang Census. The data on road network density, number of intersections, destination accessibility, and public transportation accessibility are from 2020 Quanzhou Road Network Data. The data on residential land, public management and public service land, commercial service land and transfer land are from the 2015–2035 Jinjiang City Land and Spatial Planning. POI data come from Gaode map POI data. The greening rate data come from the 2020 Quanzhou NDVI Index. 

Residential building density examines whether the vision of ASD children’s residential communities is sufficient. Residential greening examines the greening degree of ASD children’s residential communities. Destination accessibility examines how easy it is for ASD children to travel from home to their destination and whether it will be a burden on the body. Transportation service facilities examines the number of sidewalks on the travel route, the number of traffic lights, and the completeness of facilities such as separate facilities for people and vehicles. NDVI vegetation coverage examines the abundance of plant species and quantities on the travel path. Population density looks at the number of people in the destination. Number of children in the destination examines the number of children at the destination of the same age as the ASD children. Public facilities examines the number and availability of public facilities in the destination, such as public toilets. Number of recreational facilities and recreational playability examine whether children with ASD need to wait in line for their favorite facilities and whether the facilities need maintenance.

### 2.3. Anonymous Questionnaire

The anonymous questionnaire consisted of two parts: sociodemographic background information on children with ASD and children’s play behavior.

The sociodemographic background information on the children with ASD includes the local household registration, the gender of the child, the age of the child, the degree of autism, the number of children in the family, the education level of the guardian, the guardian’s employment and the monthly household income. 

Information on play behavior of children with ASD was derived from parental reports. One of the reasons why self-reporting is not used for children with ASD is that there may be language barriers that affect the accuracy of the reported results [20]. On the other hand, studies have shown that children with ASD rate their own social skills higher than their parents or teachers, making them more likely to skew reported results [21]. The answer is set in the form of a Crete five-point scale, which is subjectively evaluated by parents. The questionnaire items are all positive descriptions, such as: Can the child keep his eyes on others who are playing? The answer is: strongly disagree, disagree, neutral, agree, strongly agree (Table 4).

### 2.4. Retrospective Semi-Structured Interview

The retrospective semi-structured interview is an in-depth exploratory research method that collects original data from interviewees after the event. During the interview process, the interviewer can appropriately guide the interviewee’s conversation toward the research topic and obtain as rich data as possible.

The interviewees were selected from the parents who participated in this investigation at the special education school in Jinjiang City. Parents were invited to assist in this research by principals and teachers, and the interview was conducted by telephone. At the same time, the consent of the interviewee was obtained for recording, and the voice responses were converted into text for the record to prevent any omission of key points.

(1)The urban built environment in the interview outline

This part required parents to evaluate their satisfaction with the urban built environment, and the answers were strongly disagree, disagree, neutral, agree, strongly agree.

(2)Play behavior of children with ASD

To validate the data analysis results in the previous subsection, this section asked parents to answer whether the urban built environment promoted, inhibited or had no effect on the play behavior of children with ASD. Considering the parent’s time to answer and comprehension, this part replaces the strongly disagree, disagree, neutral, agree, strongly agree in the questionnaire with facilitation, inhibition or no influence. 

### 2.5. Data Analysis Method

The WeChat mini program was used to collect the questionnaire information, and at the end, the data from the fill-in person were extracted and sorted in the server background, and the original data were maintained for subsequent analysis.

The data from the questionnaire were analyzed in stata16SE with stepwise regression. Stepwise regression analysis introduces variables into the model one by one. After each explanatory variable is introduced, it must be tested, and the selected explanatory variables are tested one by one. When the originally introduced explanatory variables become no longer appropriate due to the introduction of later explanatory variables, if significant, delete it. This is a method that ensures that only significant variables are included in the regression equation each time a new variable is introduced. Stepwise regression analysis is a multiple regression analysis method for studying the interdependent relationships between multiple variables, while stepwise regression analysis is often used to establish an optimal or suitable regression model, so as to study the dependencies between variables more deeply.

## 3. Results

### 3.1. Analysis Results of Objective Quantitative Data

The WeChat mini program was officially launched on 15 May 2021. It took nearly 7 months, and 57 valid questionnaires were collected.

(1)The sociodemographic background characteristics of ASD children

Among the families of ASD children who participated in the survey, 51 children had a registered permanent residence in the city, accounting for 89% of the total. Among them, 47 children were boys, accounting for 82% of the total. There were 41 children aged 6–10, accounting for 72% of the total, 12 children aged 11–15, accounting for 21% of the total, and 4 children aged 0–5. There were no children aged 16–20. In terms of the degree of autism after hospital diagnosis, 36 children, or 63% of the total, were mild to moderate; 11 children were mild to asymptomatic and 10 children were severe. There were 2 children in 28 families, accounting for 49% of the total, 21 families were only-child families and 8 families had three children. There was no family with more than 3 children in this survey. In terms of education level, 22 families had guardians with college degrees, accounting for 38% of the total, 19 families had guardians with junior high school education, 9 guardians with high school education and 8 guardians with primary school education and below; only one family had a guardian with a master’s degree or above. Among them, 30 guardians were employed. In terms of income level, 21 families had a monthly income of RMB 4000 to 6000, accounting for 37% of the total; 12 families had a monthly income of RMB 6001 to RMB 9000; 10 families had a monthly income of below RMB 4000; 11 had a monthly income of more than RMB 12,000 and 3 had a monthly income of RMB 9000–12,000 (Table 5). 

(2)The impact of sociodemographic background

Play behavior is divided into five parts for description, and the five parts are assigned points and discussed together. The effect of sociodemographic background on the play behavior of children with ASD was investigated by variance analysis.

The results showed that only whether the primary guardian of the child was employed, the sex of the child, the age of the child, and the monthly household income had significant differences in the play behavior of children with ASD (Table 6). The *p*-value is 0.0680, indicating that at the significance level of 0.1, there is a significant difference in the play behavior of ASD children whether the guardian employment. The *p*-value is 0.064, indicating that at a significance level of 0.1, there is a significant difference in the avoidance behavior of ASD children depending on whether the guardian is employed. The *p*-value is 0.035, indicating that at the significance level of 0.05, there is a significant difference between the gender of the child in the concealment behavior of ASD children. The *p*-value is 0.087, indicating that at a significance level of 0.1, there is a significant difference in the age of the child in the concealment behavior of ASD children. The *p*-value is 0.074, indicating that at a significance level of 0.1, there is a significant difference between the monthly household income and the concealment behavior of ASD children. 

There were no significant differences in the level of autism, the number of children in the family, and the education level of the children’s primary guardians on the play behavior of children with ASD.

(3)The impact of the urban built environment

The questionnaire data were processed by stepwise regression analysis. The results showed that there was no significant relationship between the urban built environment and the observation behavior of children with ASD.

The number of recreational facilities had a significant positive correlation with the play behavior of children with ASD.

There was a significant positive correlation between NDVI vegetation coverage and public facilities on the avoidance behavior of children with ASD.

The number of children in the destination and excellent transportation service facilities have significant positive correlations with the concealment behavior of children with ASD.

The residential greening, public facilities and population density have significant negative correlations with the concealment behavior of children with ASD (Table 7).

### 3.2. Analysis Results of Retrospective Semi-Structured Interview

A total of 31 parents of children with ASD participated in the retrospective semi-structured interview. The interview results are shown in Table 8.

With higher residential building density, children’s play behavior is promoted.

With higher residential greening, children’s avoidance behavior and concealment behavior are inhibited.

With higher destination accessibility, children’s play behavior is promoted while concealment behavior is inhibited.

With excellent transportation service facilities, children’s observation behavior is promoted, and avoidance and concealed behavior are inhibited.

With higher NDVI vegetation coverage, children’s play behavior is promoted, while the avoidance behavior is inhibited.

With higher population density, children’s avoidance behavior and concealment behavior are promoted. 

The more children in the destination, the more children’s avoidance behavior and concealment behavior are promoted. 

With excellent public facilities, children’s avoidance behavior and concealment behavior are inhibited.

A larger number of recreational facilities and better playability, children’s observation behavior is promoted, play behavior is promoted and avoidance behavior is inhibited.

## 4. Discussion

### 4.1. The Impact of Sociodemographic Background on Play Behavior of Children with ASD

There are significant differences in the play behavior and avoidance behavior of ASD children whether the guardian is in office or not. Parents are their children’s best teachers, and the amount of time spent with them helps develop correct values and worldviews. A growing body of research shows that parental involvement or leadership in individualized therapy has a significant positive impact on the recovery of children with ASD. When a child with ASD behaves inappropriately, the guardian will promptly dissuade them and report that this is a wrong behavior. Intervention therapy promotes the generalization and maintenance effect of ASD children’s behavior through the repeated correction of stereotypical behavior. For some children with severe symptoms, the school will require guardians to accompany the children in class to stabilize their mood and behavior.

Gender and age have significant differences in the concealment behavior of children with ASD. In general, there is little difference in behavior between boys and girls with autism before the age of 5, but with age, boys showed more frequent externalizing behavior problems, such as aggression and hyperactivity, while girls showed greater internalization difficulties, such as more frequent nonverbal communication and prosocial behavior. The gender bias in the field of autism research has also resulted in relatively few studies of girls with ASD in the relevant academic literature, so the behavioral phenotype and symptoms of girls with autism remain widely unknown [22].

There is a significant difference between monthly household income and the concealment behavior of children with ASD. Parental income is an important predictor of household quality of life: children with ASD have improved quality of life with higher income. The cost of autism diagnosis and psychological counseling is high, and intervention therapy is also a considerable expense for the average family. Family income during childhood will cause individual differences in children’s subjective well-being, which will also affect children’s sense of experience and satisfaction with and cognition of things. In turn, cognition of things may cause children with ASD to be more sensitive than ordinary children and to display concealment behaviors in games [23].

### 4.2. The Impact of Urban Built Environment on Play Behavior of Children with ASD

There were many differences between the data analysis results and the retrospective semi-structured interview results. Therefore, we combined the results of the two experiments to draw the following conclusions:

The residential building density and residential greening have influence on the play behavior of children with ASD. Since the home is the place where they spend the most time, this means that both the physical and cultural environment of the residential area have long-term impacts on children with ASD. In addition to bringing visual oppression, higher building density also generates more occupants. Although there are some cases of insensitivity to crowd density, this can cause psychological stress for most children with ASD [24]. A higher greening rate in residential areas can promote play behavior in children with ASD. On the one hand, many researchers have confirmed that a high greening rate can relieve the nervousness of ASD children and play a certain role in soothing. In particular when children with ASD have hidden behaviors, the shade and cover of green plants can provide them with good shelter [25]. On the other hand, green plants also provide a medium for children to explore the world. Many ASD children have a surprising interest in plants. This topical interest can also play a mediating role in getting along with peers. 

Destination accessibility has influence on the play behavior of children with ASD. Higher accessibility means reasonable distances and travel time without creating a physical burden and losing interest in games. Areas with high accessibility generally have urban built environments that attract people to walk, such as rich business varieties and excellent facilities. Children with ASD themselves have limited interests, and if the urban built environment remains unchanged during travel, it will easily give them a dull and boring psychological feeling. If this emotion is not relieved after reaching the destination, it will affect their play behavior [26]. 

Transportation service facilities influence the play behavior of children with ASD. Reasonable sidewalk settings can not only effectively control the speed of vehicles and improve the safety perception of children with ASD on the environment but also regulate their stereotyped behavior and improve the safety travel. The people–vehicle isolation facility composed of green belts can visually weaken the edge of the road and play a soothing role for children with ASD. Less on-street parking can make the street look wider, and the expanded field of view can help ease anxiety [27].

NDVI vegetation coverage has influence on the play behavior of children with ASD. Being in a green environment provides more opportunities for children to relax and unwind. Compared with ordinary children, children with ASD are more able to obtain this benefit from green plants. Green plants can increase activity in children with ASD, which in turn increases the possibility of play. Greenery can also provide a shelter when children with ASD are socially frustrated [28].

The population density of the destination and the number of children at the destination have influence on the play behavior of children with ASD. Under normal circumstances, children with ASD will have anxiety in places with high population density, which may easily lead to the occurrence of avoidance and concealment behaviors. However, there is also a case where the influencing behavior is not related to the population density of the destination but to the number of children at the destination [29]. This is also the reason for the interest orientation of children with ASD themselves, they seem to be able to focus only on the things they like. Things that do not interest them generally will not have much impact if they do not act directly on them. As the study found, a girl can play autonomously in a crowded park but withdraw when older boys approach her. In this study, from the first discomfort, the girl became fearful and withdrawn and finally quit the game she was interested in. 

Public facilities have influence on the play behavior of children with ASD. For example, the setting of park benches was originally used for parents to rest, but different functional blocks are divided in space planning [30]. The uncertainty of children with ASD themselves make them have a certain chance of being out of the current situation in the game. This result may be the biased analysis results due to the small sample size on the one hand and the insufficient classification of public facilities on the other hand. Some public facilities, such as restrooms and rest facilities, are caused by differences in judgment. This is also the direction that follow-up research will continue to work on. 

The number and playability of recreational facilities have influence on the play behavior of children with ASD. The main purpose of children is to play, so the variety of recreational facilities and a certain number of recreational facilities of the same kind will have an important impact on the games of children with ASD. The number of recreational facilities determines the waiting time, and the playability of recreational facilities determines whether children with ASD are interested [31]. However, during the research, it was also found that only some children with ASD played fixed recreational facilities every time and did not show interest in other facilities. 

The results of data analysis showed that the social context made significant differences in the children’s play behavior. Although the subject of this study is the impact of the urban built environment on play behaviors in children with ASD, social context also plays a role that cannot be underestimated. For example, the level of household income may affect the location of the residence. Therefore, this part discusses the influence of family background on the play behavior of ASD children to supplement the research conclusions and enrich the conclusions.

## 5. Conclusions

At present, the research on the urban built environment and ASD children’s play behavior is almost blank. It may be that there are so many uncertainties in children with ASD that research is difficult to conduct or that it is not easy to obtain research samples and most of the related studies are case studies. This study constructs the relationship between the play behavior of children with ASD and the urban built environment, explores the influencing factors of the urban built environment on the play behavior of children with ASD and achieves certain results.

We found that there were significant differences in the play behavior and avoidance behavior of ASD children depending on whether the guardian was employed. Gender and age had significant differences in the concealment behavior of children with ASD. There was significant association between the monthly household income and the concealment behavior of children with ASD. In addition, lower residential building density, higher residential greening and higher destination accessibility had positive effects on ASD children’s play behavior. Excellent transportation service facilities and higher NDVI vegetation coverage have positive effects on the play behavior of children with ASD. More recreational facilities and recreational playability have positive impacts on the game behavior of children with ASD. The population density and number of children in the destination, as well as public facilities, may influence the play behavior of children with ASD.

Therefore, how to make ASD children benefit from games as much as possible in the limited time outside, such as relieving anxiety and improving social skills, has become an important issue for parents to consider. In view of this, based on the comprehensive consideration of sociodemographic background and the impact of urban built environment on ASD children’s play behavior, we make the following recommendations: (1)When choosing a place to live, in addition to the building density of the residential area, the degree of greening should also be considered because low building density and high greening can not only promote the physical and mental health of children with ASD, but also help their play behavior.(2)In the choice of travel path, a road with more sidewalks, more traffic lights and higher vehicle speed limit should be selected as much as possible. At the same time, effective isolation facilities for people and vehicles are also very important. Parents should try to choose a road close to nature, which will play a positive role in promoting the play behavior of children with ASD after arriving at the destination.(3)In the choice of destination, green space with a larger number of recreational facilities should be preferred. The crowd at the destination is also important, and children-oriented parks should be considered first. Secondly, the sense of security is also an important reference for children with ASD. The monitoring functions of the park itself, such as patrolling personnel, lighting equipment and even monitoring from surrounding crowds are also important. Whether there are noise, odor and excellent public facilities will also have positive effects on the play behavior of children with ASD. The type and quantity of recreational facilities is also an important reference. There are more recreational facilities of the same type, which can reduce waiting time. The rich variety of recreational facilities can provide more choices. Some recreational facilities that require collaboration can promote the play behavior of children with ASD.

However, due to the particularity of the research subjects, the sample size of the two experiments gradually decreased. This is partly due to conceptual issues that parents do not want their children to be used as research samples. On the other hand, some users have compatibility problems with WeChat mini programs, resulting in a small sample size. At the same time, due to the mobile phone network, the positioning information cannot be updated, resulting in missing samples.

At this stage, it can only be concluded that the urban built environment has an impact on the play behavior of children with ASD. In future research, other methods and tools will be tried to further explore the impact mechanism. It is hoped that the results of this research can enrich the content of ASD children’s play behavior in related fields and provide a new perspective for planning and building a child-friendly city. At the same time, it also pays attention to social disadvantaged groups, using children with ASD as a medium, pays attention to social justice and explores the new direction of urban sustainable development.

## Figures and Tables

**Figure 1 ijerph-19-14752-f001:**
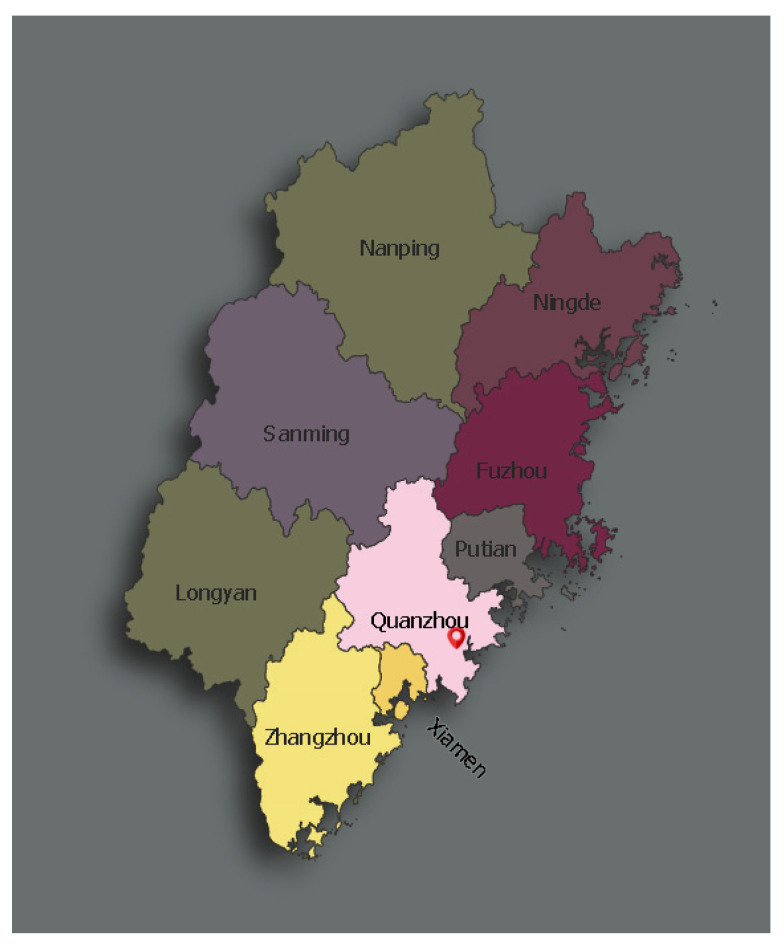
The special education school in Jinjiang City, Fujian Province.

**Figure 2 ijerph-19-14752-f002:**
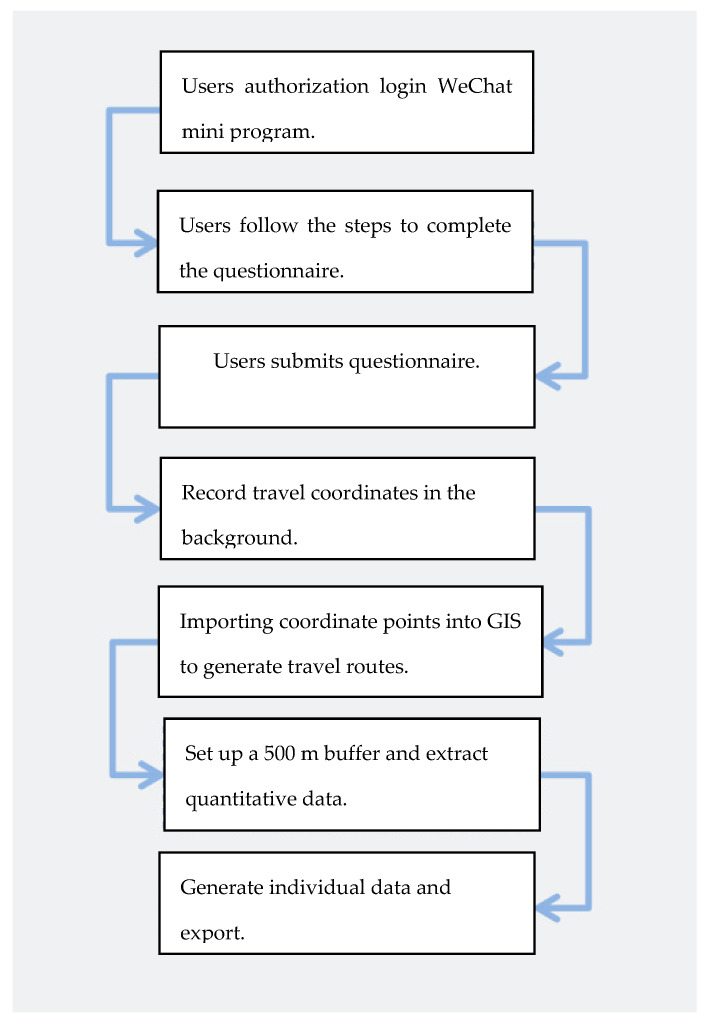
WeChat mini program collects the steps of urban built environment.

**Table 1 ijerph-19-14752-t001:** Autism Diagnostic Scale.

Key Words	Describe
eye avoidance	Avoid other people’s eyes or look directly into the face when playing or communicating with others.
social smile	Smile socially at others in appropriate situations or social situations.
understand gestures and facial expressions	Understand the gestures and facial expressions of others in social situations.
active display	Actively show toys to others during play.
sharing	Share food, toys, or other items with others.
sympathy	Show sympathy to someone who is showing sadness, hurt, or illness.
common concern	When others point or touch objects, accommodate others’ shared concerns.
expression management	Appropriate facial expressions.
others approach	Do not react negatively to the approach of others.
help others	Offer to help others when they need it.
attract attention	Can get the attention of others in the right way.

**Table 2 ijerph-19-14752-t002:** Autism Children’s Development Assessment Scale.

Key Words	Describe
non-verbal skills in social	gaze at social objects.
	physical contact within 3 m of others.
	allow others to walk around.
	sit quietly.
social skills	respond to others with a smile.
	trigger a response by smiling or making a sound.
	ask for help, get what he or she wants.
	reacting to unfamiliar situations or to strangers
	share with others.
Social etiquette	respond to someone’s greeting with a smile.
	reaching out, holding or shaking hands in response to someone’s greeting.
	wave goodbye.

**Table 3 ijerph-19-14752-t003:** General steps for play behavior in children with ASD.

Play Behavior	Describe
Observation (Obs)	Children can keep their eyes on others who are playing.
Participation (Par)	Children can respond to the actions of others in their own way, such as shout loudly or showing an intent to participate.
	Children can fit in naturally or intervene strongly (snatching behavior) in the play of others.
	Children can accept invitations from others to participate in the game.
	Children can seek help from others to participate in play.
	Children can help others (pass items) during play.
Avoidance (Avo)	Children take the initiative to withdraw from games that they are unable to do or are not interested in.
Concealment (Con)	After the child quits the game, when others approach, seek a hidden space or the arms of the parents to hide themselves.

**Table 4 ijerph-19-14752-t004:** Anonymous Questionnaire Details.

	Question	Answer
Sociodemographic background information of children with ASD	Have a local household registration?	Yes/No
	The gender of the child?	Male/Female
	The age of the child?	0–5 years old/6–10 years old/11–15 years old/16 years old-20 years old
	The degree of autism?	Mild to asymptomatic/Mild to moderate/Severe
	The number of children in the family?	1 child/2 children/3 children/more than 3 children
	The education level of the guardian?	primary and below/junior high school/high school/university/master and above
	The guardian employment?	Yes/No
	The monthly household income?	Below RMB 4000, RMB 4000–6000, RMB 6000–9000, RMB 9000–12000, RMB 12,000 or more
Children’s play behavior	Children can keep their eyes on others who are playing?	Strongly Disagree, Disagree, Neutral, Agree, Strongly Agree
	Children can respond to the actions of others in their own way, such as shout loudly or showing an intent to participate?	Strongly Disagree, Disagree, Neutral, Agree, Strongly Agree
	Children can fit in naturally or intervene strongly (snatching behavior) in the play of others?	Strongly Disagree, Disagree, Neutral, Agree, Strongly Agree
	Children can accept invitations from others to participate in the game?	Strongly Disagree, Disagree, Neutral, Agree, Strongly Agree
	Children can seek help from others to participate in play?	Strongly Disagree, Disagree, Neutral, Agree, Strongly Agree
	Children can help others (pass items) during play?	Strongly Disagree, Disagree, Neutral, Agree, Strongly Agree
	Children take the initiative to withdraw from games that they are unable to do or are not interested in?	Strongly Disagree, Disagree, Neutral, Agree, Strongly Agree
	After the child quits the game, when others approach, seek a hidden space or the arms of the parents to hide themselves?	Strongly Disagree, Disagree, Neutral, Agree, Strongly Agree

**Table 5 ijerph-19-14752-t005:** The sociodemographic background characteristics of ASD children.

Sociodemographic Background	Result (%)
Have a local household registration?	Yes (89%)/No (11%)
The gender of the child?	Male (82%)/Female (18%)
The age of the child?	0–5 years old (7%)/6–10 years old (72%)/11–15 years old (21%)/16 years old-20 years old (0%)
The degree of autism?	Mild to asymptomatic (19%)/Mild to moderate (63%)/Severe (18%)
The number of children in the family?	1 child (37%)/2 children (49%)/3 children (14%)/more than 3 children (0%)
The education level of the guardian?	primary and below (14%)/junior high school (30%)/High school (16%)/University (38%)/Master and above (2%)
The guardian employment?	Yes (51%)/No (49%)
The monthly household income?	Below 4000 RMB (18%)/4000–6000 RMB (37%)/6000–9000 RMB (21%)/9000–12000 RMB (5%)/12,000 RMB or more (19%)

**Table 6 ijerph-19-14752-t006:** The impact of sociodemographic background on play behavior of children with ASD.

Play Behavior	Sociodemographic Background	*p*
Participation (Par)	guardian employment	0.068
Avoidance (Avo)	guardian employment	0.065
Concealment (Con)	the gender of the child	0.039
	the age of the child	0.088
	the monthly household income	0.074

**Table 7 ijerph-19-14752-t007:** The impact of urban built environment on play behavior of children with ASD.

	Play Behavior			
	Obs	Par	Avo	Con
Residential building density				
Residential greening				−5.158 ***
Destination accessibility				
Transportation service facilities				0.093 **
NDVI vegetation coverage			4.097 *	
Population density				−0.052 *
Number of children in the destination				0.573 ***
Public facilities			0.114 *	−0.078 **
Number of recreational facilities		2.180 **		
Recreational playability				
Constant	1.273 ***	1.182 **	1.739 ***	1.283 **

Observations				
R–squared	0.642	0.609	0.688	0.702
Adj R–square	0.629	0.588	0.669	0.685

*** *p* < 0.01, ** *p* < 0.05, * *p* < 0.1.

**Table 8 ijerph-19-14752-t008:** Results of retrospective semi-structured interview.

Urban Built Environment	Play Behavior of Children with ASD											
	Obs			Par			Avo			Con		
	P	I	N	P	I	N	P	I	N	P	I	N
Residential building density					21	10						
Residential greening								26	5		19	12
Destination accessibility				28		3					21	10
Transportation service facilities	26		5					27	4		29	2
NDVI vegetation coverage				31				29	2			
Population density							19		12	20		11
Number of children in the destination							20		11	21		10
Public facilities								26	5		22	9
Number of recreational facilities	31			29		2		20	11		26	5
Recreational playability	20		11	25		6		24	7		27	4

P = promotes, I = inhibits, N = no effect. The numbers represent the number of people who chose the option.

## Data Availability

Not applicable.

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
