# Peer review of "The Impact of the Urban Built Environment on the Play Behavior of Children with ASD"

_ijerph, 2022, doi:10.3390/ijerph192214752_

Round 1

Reviewer 1 Report

Comments

This article studies the impact of urban built environment on children with ASD, which is more meaningful and very helpful for the care of children with ASD; after reading the article, there are still some problems with the article; therefore, I offer some comments.

Introduction

1 The introduction is very simple, and the logic is confusing. The author is requested to carefully revise the content of the introduction part; especially the research background.

2 The second paragraph of the introduction mentions the relationship relationship between the urban built environment and children's play, but this is very small and insufficient to support the overall article; specifically, the author should list more literature, mainly on ASD children and cities. relationship to the environment.

3 It is recommended that the authors add the theoretical basis for influencing the activity behavior of children with ASD so that readers cold be more specific about children with ASD.

4 At lines 74-75, the author mentions ASD children for Chinese conditions, but lacks relevant descriptions; it is recommended that the author refer to relevant materials to supply the information on ASD children about Chinese conditions, and then highlight this research.

5 The authors should identify the innovations of this paper clearly, introduce the importance of research on the urban built environment and children with ASD in the introduction, and explain why this research is conducted.

Methodology

6 Lines 86-87, the author mentioned the study site, please consider whether to add a location map

7 The urban built environment is an important indicator in this study, at lines 115-117, the author mentions the use of the WeChat app to collect the urban built environment. It is suggested that the author add a new sub-heading to introduce the built environment of the city, for example: 2.3 The urban built environment, and also add a guide figure explaining how to collect the urban environment index.

8 Line 99, the author is asked to show the details of the questionnaire

9 Please add the flowchart for the author of this article

Results

10 At lines 152-154, “The results showed that only whether the primary guardian of the child was employed, the sex of the child, the age of the child, and the monthly household income had 153 significant differences in the gaming behavior of children with ASD (Table 2). But it's hard to understand; what does the “P” of Table 2 stand for? What is the relationship between the results of the study and “P”? There is no explanation in paper, the author please add relevant explanations.

11 At line 160, regression analysis was mentionsed, which is a research method; but the author did not introduce this method in the Methodology of Chapter 2.

12 At line 165, the NDVI index appears; as far as I know, NDVI is the landscape index, which can also describe the built environment of the city; why does NDVI appear suddenly? The author should consider whether to add relevant explanations in the chapter of Methodology.

13 There are many indexes related to the urban built environment in Table 3, but there are no explanations and introduction in the paper. The author is requested to add relevant explanations because the urban built environment is an important index of this research.

Discussion

14 At lines 229-295, the author discusses the relationship between different urban built environment factors and children with ASD, and also done a lot of work; however, it is difficult to find the research focus; please summarize the content of this chapter, highlight the research focus, achieve the value of this research.

15 The title of this study is: The impact of urban built environment on play behavior of children with ASD; but in the discussion section, the author discusses the impact of social context and urban built environment on children with ASD, please explain why?

Conclusion

16 It is recommended that the author change the title “conclusion” to “conclusion and suggestion”,

17 Please describe the specific content of the conclusions of this paper clearly, as this chapter does not seem to see the actual conclusions of the research

18 The author mentions three suggestions, which are very meaningful; however, the suggestions are based on research findings and results; besides, your suggestion is missing a link relationship to this article conclusions, please add relevant explanations. And note that conclusions and results are not the same.

19 Lines 301-303, the author mentions “explores the influencing factors of the urban built environment on the play behavior of children with ASD during the COVID-19 pandemic, and achieves certain results”; however, the content of discussing COVLD-19 is on line 304, please modify it. Also, consider whether to put this content in the Discussion section, because it belongs to discussion.

Author Response

Dear reviewer,

Thank you for your valuable advice, let me benefit a lot!

Attached is my reply to your suggestion, please check!

Best regards,

Shengzhen Wu

Author Response

(The authors gave the same response as above.)

Author Response

(The authors gave the same response as above.)

Round 2

Reviewer 1 Report

In my opinion, the authors have done many work and made a lot of revisions; I think the current paper could be published.

Author Response

Dear editor,

I appreciate the positive feedback from you. Thank you so mush!

sincerely Yours'

Shengzhen Wu